behaviour/ecology

*Arctocephalus*, central-place foraging (CPF), El Niño Southern Oscillation (ENSO), Indian Ocean Dipole (IOD), marine predator, otariid

**Author for correspondence:**
Cassie N. Speakman
e-mail: cspeakman@deakin.edu.au

# Influence of environmental variation on spatial distribution and habitat-use in a benthic foraging marine predator

Cassie N. Speakman[1], Andrew J. Hoskins[2], Mark A. Hindell[3], Daniel P. Costa[4], Jason R. Hartog[5], Alistair J. Hobday[5] and John P. Y. Arnould[1]

[1]School of Life and Environmental Sciences, Deakin University, Burwood, Victoria, Australia
[2]CSIRO Health and Biosecurity, Townsville, Queensland, Australia
[3]Institute for Marine and Antarctic Studies, University of Tasmania, Hobart, Tasmania, Australia
[4]Ecology and Evolutionary Biology Department, University of California Santa Cruz, Santa Cruz, CA, USA
[5]CSIRO Oceans and Atmosphere, Hobart, Tasmania, Australia

CNS, 0000-0002-0023-518X; AJH, 0000-0001-8907-6682;
JPYA, 0000-0003-1124-9330

The highly dynamic nature of the marine environment can have a substantial influence on the foraging behaviour and spatial distribution of marine predators, particularly in pelagic marine systems. However, knowledge of the susceptibility of benthic marine predators to environmental variability is limited. This study investigated the influence of local-scale environmental conditions and large-scale climate indices on the spatial distribution and habitat use in the benthic foraging Australian fur seal (*Arctocephalus pusillus doriferus*; AUFS). Female AUFS provisioning pups were instrumented with GPS or ARGOS platform terminal transmitter tags during the austral winters of 2001–2019 at Kanowna Island, south-eastern Australia. Individuals were most susceptible to changes in the Southern Oscillation Index that measures the strength of the El Niño Southern Oscillation, with larger foraging ranges, greater distances travelled and more dispersed movement associated with 1-yr lagged La Niña-like conditions. Additionally, the total distance travelled was negatively correlated with the current year sea surface temperature and 1-yr lagged Indian Ocean Dipole, and positively correlated with 1-yr lagged chlorophyll-*a* concentration. These results suggest that

environment variation may influence the spatial distribution and availability of prey, even within benthic marine systems.

## 1. Introduction

The marine environment is highly dynamic, with substantial variation at multiple spatial and temporal scales [1,2]. Much of this variability has been linked to large-scale climate processes, such as the El Niño Southern Oscillation (ENSO) and the Pacific Decadal Oscillation [3,4], that strongly influence the conditions experienced at the local scale. In pelagic systems, environmental change leads to shifts in prey distribution [5,6] and productivity [7,8], with current activity dictating where prey resources will congregate.

Central-place foraging (CPF) marine predators are constrained in their foraging range and duration due to offspring provisioning requirements [9], making them susceptible to environmental variability. If prey distributions shift, CPF individuals need to increase their foraging effort to obtain enough energy for personal maintenance and offspring provisioning [10–12]. The higher costs to CPF marine predators under reduced prey availability or accessibility can have significant impacts on their reproductive success [13,14]. This is particularly concerning as the balance of phases of large-scale climate drivers are predicted to change over the coming decades [15–18], with substantial consequences for the distribution and abundance of pelagic species. However, benthic systems are considered more stable and, as such, benthic CPF predators have been assumed to have lower susceptibility to environmental variability [19].

Climate-mediated changes at lower trophic levels can have considerable influence on higher trophic levels, resulting in changes in marine predators' habitat use (e.g. [20]), diet (e.g. [21,22]) and foraging behaviour (e.g. [23–25]). However, many of the studies investigating climate-mediated effects on marine predators have been focused on pelagic foraging predators, with relatively little currently known about how benthic predators are likely to be impacted under future environmental change. While pelagic marine environments are known to be highly spatio-temporally variable, with high levels of marine primary productivity and species abundance [26], less in known about the temporal variability of benthic habitat communities and, thus, the influence of future environmental change on benthic communities or the benthic marine predators that depend on them. Such knowledge is crucial for predicting how such benthic communities may respond to anticipated environmental change.

Australian fur seals (*Arctocephalus pusillus doriferus*; AUFS) are predominantly benthic foragers, hunting almost exclusively on the seafloor of Bass Strait [27,28] on a wide variety of prey types (greater than 60 species) [29–31]. While annual pup production is estimated at only *ca* 28–47% of pre-sealing levels, AUFS represent the largest marine predator biomass in south-eastern Australia [32,33]. Like other otariids, females adopt a CPF strategy during lactation [34], undertaking trips at sea for 2–11 days before returning to provision pups [27,28]. Consequently, female AUFS may be particularly vulnerable to changes in prey availability and distribution. Despite the more stable nature of the benthic marine environment, environmental variability has already been shown to impact the foraging behaviour and diet of female AUFS [24,31,35]. However, the impacts on habitat use, which can have substantial consequences for the reproductive success of CPF predators, are unknown.

The south-east Australian marine region is experiencing rapid environmental change [36,37], with further changes anticipated over the coming decades [38]. The region is strongly influenced by three main climate drivers: the Indian Ocean Dipole (IOD) [39]; the Southern Annular Mode (SAM) [40]; and the ENSO [4]. The phases of these climate drivers are associated with changes in sea surface temperature (SST), sea surface height (an indicator of eddy activity), zonal wind strength and primary productivity in the region [4,40]. The fairly uniform bathymetry and typically low marine primary production [41] in the shallow (60–80 m) continental shelf of Bass Strait, south-eastern Australia, make the region an ideal study area to investigate how benthic predators are influenced by perturbations in their environment. The region is influenced by three separate water bodies: the South Australia Current (SAC), driving warmer water along with the southern coast into Bass Strait; the Sub-Antarctic Surface Waters (SASW), bringing cool, nutrient-rich waters into Bass Strait, and mixing with the SAC waters; and the nutrient-poor East Australian Current (EAC). The influence of these water bodies on the primary productivity and, hence, pelagic prey distribution within Bass Strait is largely driven by large-scale climate conditions [40,42]. While it is known that environmental variation has consequences for pelagic CPF predators through changes in prey distribution, it is important to determine how environmental change impacts benthic marine predators.

The aims of the present study, therefore, were to examine in female AUFS: (i) the interannual variation in spatial distribution and habitat use and (ii) the local- and broad-scale factors influencing this variation. Such information is important for determining how the species may be impacted by

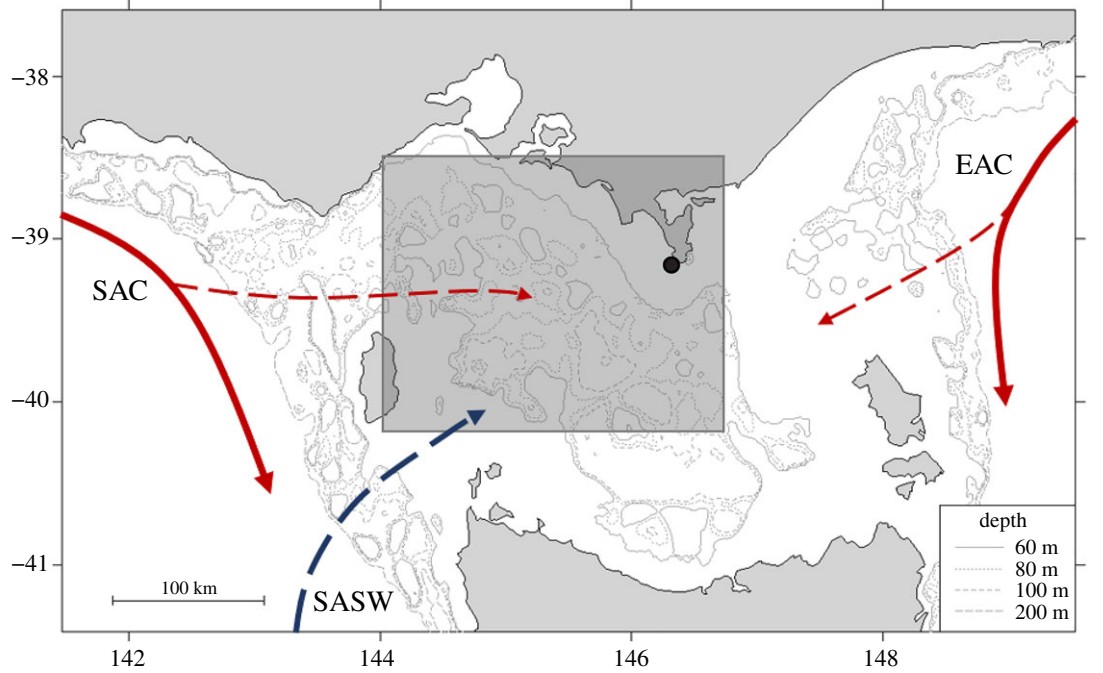

**Figure 1.** Location of the Kanowna Island breeding colony (black circle) within south-eastern Australia and inflow of major water bodies (SAC—South Australian Current; SASW—Sub-Antarctic Surface Waters; EAC—East Australian Current) into Bass Strait. Solid lines represent current flow and dashed lines represent water flow into Bass Strait. Red lines indicate warm water and blue lines represent cool water. The shaded box indicates the region for which local-scale environmental conditions were derived, which encompasses the main foraging area of female AUFS.

future environmental change and shifts in prey availability, while increasing our understanding of the sensitivity of benthic marine predators to environmental change.

## 2. Methods

### 2.1. Animal handling and instrumentation

The study was conducted at Kanowna Island (39°10′ S, 146°18′ E; figure 1) in northern Bass Strait, south-eastern Australia. Sampling occurred during the austral winter (April–August), the period of peak nutritional demand for lactating females [27], each year between 2001–2003 and 2006–2019. The island hosts the third largest breeding colony of AUFS with an annual pup production of *ca* 3400 [32,33]. Breeding areas are spread around the island but many of them are not suitable for animal captures due to cliffy terrain [43]. The main colony at the northern end of the island, which accounts for greater than 65% of pup production [44], and its hinterland are the only suitably safe areas for the capture of adult females.

Following the pupping period, adult females mostly nurse their pups in the colony hinterland areas as they provide protection from strong winds and wave action, and the use of these areas varies with weather conditions [45]. To minimize disturbance to the colony, females observed nursing pups in these hinterland areas were selected at random for capture. Patterns of colony attendance by fur seals can be influenced by a variety of factors, including prey availability/distribution, maternal condition, pup age/condition and weather [27,46]. In addition, the timing and duration of sampling sessions were dependent on weather and logistics. Consequently, captured individuals are likely to be unbiased representatives of the lactating female population.

Individuals were captured using a modified hoop-net (Fuhrman Diversified, Seabrook, TX, USA) and anaesthetized using isofluorane delivered via a portable gas vaporizer (Stinger, Advanced Anaesthesia Specialists, Gladesville, NSW, Australia) prior to processing. Individuals were then instrumented with either an ARGOS platform terminal transmitter (PTT; Kiwisat100, Sirtrack Ltd, Havelock North, New Zealand) or GPS (FastLoc 1, Sirtrack Ltd, Havelock North, New Zealand or Mk10F, Wildlife Computers Ltd, Redmond, WA, USA). Individuals were also fitted with a VHF transmitter (Sirtrack

Ltd, Havelock North, New Zealand) to assist in relocating the animal for recapture. Devices were glued in series along with the midline dorsal pelage, just posterior to the scapula, using quick-setting two-part epoxy (RS Components, Corby, UK). The PTT devices were programmed to transmit a pulse signal every 45 s when at the surface. Satellite location data were obtained through the CLS Argos service. The GPS devices were programmed to record location at 10 or 15 min intervals when individuals were at the surface.

To aid identification, individual numbered plastic tags (Super Tags, Dalton, Woolgoolga, Australia) were then inserted into the trailing edge of each fore flipper before the individual was allowed to recover from anaesthesia and resume normal behaviour. Following at least one foraging trip to sea, individuals were recaptured as previously described and devices were removed by cutting the fur beneath the device with a scalpel blade.

## 2.2. Data processing

Location data (both PTT and GPS) were first filtered using a basic speed filter with a maximum swim speed of 6 m s$^{-1}$ to remove erroneous locations and at-sea movement tracks were linearly interpolated every 10 min using the *trip* package [47,48] within the R statistical environment [49]. A 1 km buffer around all known haul-out sites within Bass Strait was used to account for individuals resting at haul-out locations away from Kanowna Island [46]. All PTT/GPS locations (at sea and on land) occurring within these buffer zones were excluded from further analyses. A foraging trip was defined by an individual leaving and returning to Kanowna Island, with the trip duration being the time away from the colony minus any time spent at other haul-out locations. Because AUFS have been observed to spend several hours at a time in the water surrounding the colony for purposes other than foraging (e.g. thermoregulation [28]), foraging trips were defined as continuous periods of greater than or equal to 6 h in the water in which at least one foraging dive occurred.

For each foraging trip of each individual, the total trip duration (h), total horizontal distance travelled (km), mean bearing (°) and bearing to the most distal location (°) were calculated. Additionally, 50% and 95% home range estimates were calculated for each trip using kernel utilization distributions (KUD) within the *adehabitatHR* package [50]. Both home range estimates were calculated with the kernel smoothing parameter $h$ set to 'reference bandwidth' and 'least-square cross-validation'. However, as the outputs of the two smoothing parameters were highly correlated ($r > 0.8$), the home range estimates using the 'reference bandwidth' were used.

## 2.3. Environmental variables

To investigate environmental influences on habitat and space use in female AUFS, standardized monthly means of large-scale climate indices and local-scale (i.e. within the central Bass Strait region; figure 1) environmental conditions, with known or potential impacts on the prey availability for marine predators within Bass Strait, were selected for analysis [4,40,42,51,52]. At the local scale, the mean SST, sea surface chlorophyll-*a* concentration (chl-*a*), zonal (westerly) wind component and sea surface height anomaly (SSHa) were obtained as mean monthly values using the Spatial Dynamics Ocean Data Explorer (Hartog & Hobday) [53] and averaged for the austral winter (June–August) and austral spring (September–November). SSTs were derived from CSIRO three-day composite SST (1999–2008) (from https://www.marine.csiro.au/remotesensing/) and RAMSSA (2009–2019) [54]. Chl-*a* levels were derived from SeaWiFS (1999–2010) [55] and MODIS (2011–2019) [56] NASA satellite-based ocean colour imagery. Zonal wind component was extracted from NCEP reanalysis-derived data provided by the NOAA/OAR/ESRL PSL (Boulder, CO, USA) from https://psl.noaa.gov/data/gridded/data. ncep.reanalysis.derived.html [57], and SSHa from synTS [58]. All local-scale environmental variables were extracted at 4–9 km resolution.

Large-scale climate indices, including the IOD mode, SAM and the Southern Oscillation Index (SOI), were obtained as monthly values from the National Oceanic and Atmospheric Administration (https://psl.noaa.gov). Monthly values were then averaged to create annual values. These large-scale indices have been found to influence the foraging behaviour of female AUFS [24] and other pinniped species (e.g. [25,59,60]). Additionally, large-scale climate conditions can influence primary productivity and prey distribution in Bass Strait [6,15,40,61,62].

Lagged influences of climate conditions on the foraging behaviour of female AUFS have previously been reported [24,35]. These lagged effects have been suggested to occur due to changes in the recruitment of prey species, which have a delayed influence within marine food chains. Hence, to

investigate the potential influence of lagged conditions on the spatial distribution and habitat use of AUFS, 1- and 2-yr lagged conditions were included in the analyses as 1- and 2-yr lagged means.

## 2.4. Statistical analysis

All statistical analyses were conducted in the R statistical environment (version 3.6.3) [49]. Data exploration was conducted following the protocols described in Zuur *et al.* [63]. Prior to analyses, covariates were assessed for collinearity and, where $r > 0.7$ or $r < -0.7$ [64], the most biologically relevant term was retained. This led to the exclusion of the 50% home range estimate, trip duration, maximum range and distal bearing from further analyses.

Hierarchical generalized linear models (GLMs) were implemented using the *brms* package [65,66] for the 95% home range estimate and total distance travelled. Models were fitted with a Gamma distribution with log link function and ran over four Monte Carlo Markov chains with 10 000 iterations (including 3000 warm-up iterations) per chain. A random intercept effect of individual fur seal was included in all home range and total distance models. A circular GLM (circGLM) was constructed for mean bearing from colony using the *circglmbayes* package [67] with 10 000 iterations per model. To account for individual differences in behaviour and sample size, data were aggregated at the individual level for the circGLM models.

Candidate models were constructed for each response variable, including the null model, current year model (excluding lags), current and 1-yr lag model (excluding 2-yr lags), only lags model (excluding current year), 1-yr lag model, 2-yr lag model and the maximal model (all explanatory variables). Additionally, reduced models were constructed by successively removing non-significant variables from the most supported (using the methods below) candidate model until all explanatory variables were significant. To avoid over-parameterization, models were fitted for local- and large-scale variables separately.

Model selection was conducted using the 'IC_compare.circGLM' function (in the *circglmbayes* package) for the mean bearing from the colony, with the model with the lowest Watanabe–Akaike information criterion determining the best model. Candidate 95% home range estimate and total distance travelled models were compared using leave-one-out cross-validation to estimate pointwise out-of-sample prediction accuracy. The posterior odds and parameter estimates were then extracted from the optimal model. Unless stated otherwise, data are presented as median and interquartile range (IQR).

# 3. Results

A total of 110 individuals were deployed with devices between 2001–2003 and 2006–2019. Deployments ranged from 2.7–140.5 d, with an average of $3.8 \pm 3.4$ foraging trips per individual (electronic supplementary material, table S1). The spatial distribution and habitat use of female AUFS varied considerably between years (figure 2 and table 1). Foraging trips ranged from 0.3 to 28.3 d, with a median of 4.9 d (IQR: 4.4 d). Individuals had a median total distance travelled of 331.7 km (IQR: 291.0 km). The median 95% home range was 1664.9 km$^2$ (IQR: 3013.2 km$^2$) and the maximum distance travelled to the most distal point ranged from 1.7 km to 418.8 km (median: 87.3 km; IQR: 69.3 km). The mean bearing from the breeding colony was 191.5° (circular s.d.: 60.3). The annual foraging trip metrics are summarized in table 1.

## 3.1. Local-scale environmental influences on spatial distribution and habitat use

During the study period, there was considerable interannual and seasonal variation in the local-scale conditions observed in the central Bass Strait region (electronic supplementary material, table S2). Chl-*a* was lowest in 2005 (0.48 mg m$^{-3}$) and 2014 (0.38 mg m$^{-3}$), and highest in 2011 (0.85 mg m$^{-3}$) and 2019 (0.77 mg m$^{-3}$), for winter and spring, respectively. Winter SST had a low of 12.6°C in 1998 and a high of 14.2°C in 2014, while spring SST was lowest in 2003 (13.1°C) and highest in 2013 (14.1°C). Wind-*u* ranged from 2.7 to 5.9 m s$^{-1}$ and from 2.0 to 4.6 m s$^{-1}$ in the winter and spring, respectively. Additionally, the winter SSHa ranged from 0.02 to 0.09 m, while the spring SSHa ranged from −0.08 to 0.03 m.

The null model was the optimal local-scale model explaining both the variation in the 95% home range size estimate and the variation in mean bearing for female AUFS (electronic supplementary

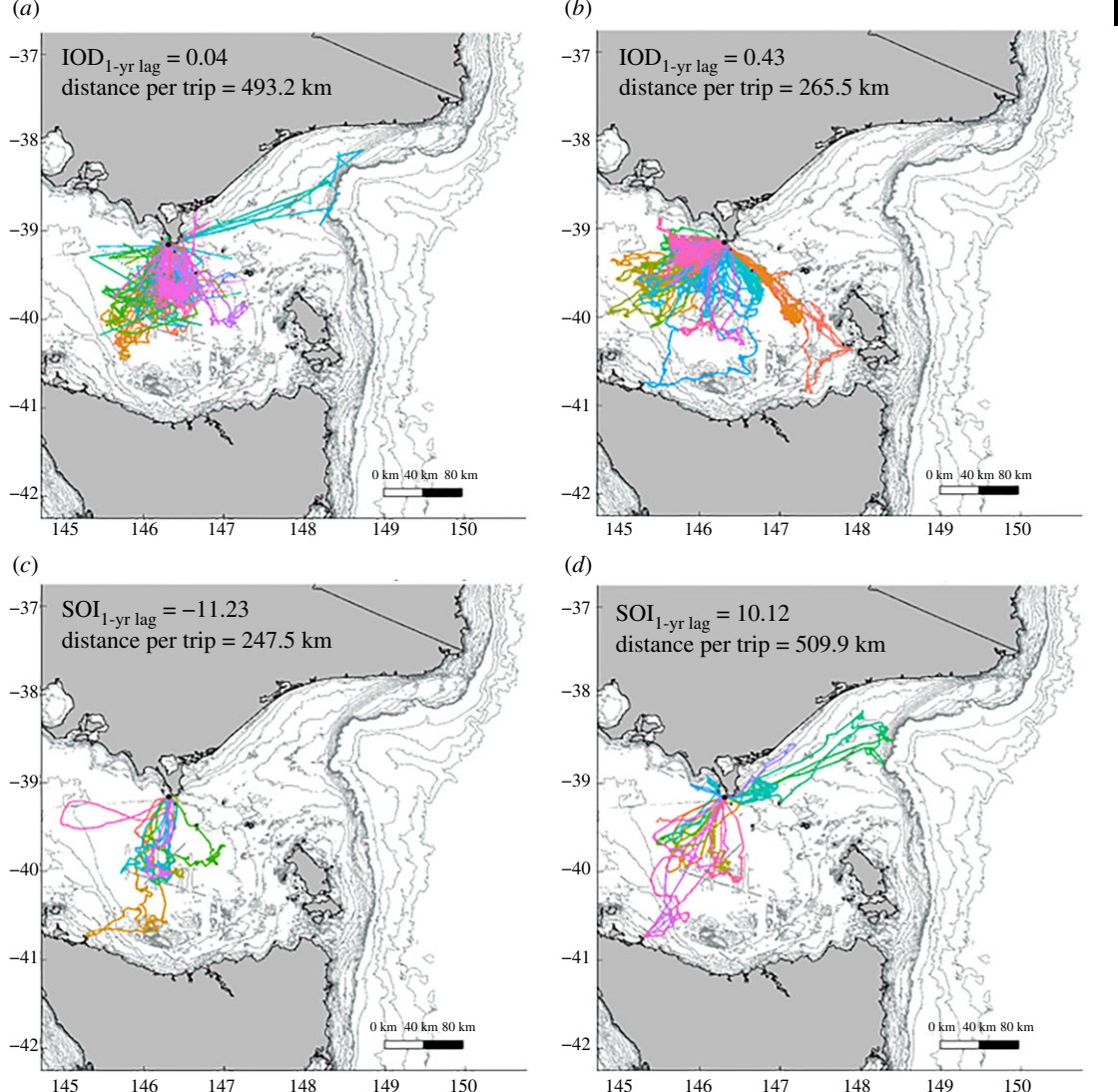

**Figure 2.** Representative tracks showing the variability in spatial distribution and habitat use of female AUFS during the austral winter, illustrating the reduction in total distance travelled with more positive 1-yr lagged IOD values ((*a*) mid/late-lactation 2006 (*N* = 6) and (*b*) mid/late-lactation 2019 (*N* = 7)) and the increase in total distance travelled and variation in mean bearing with more positive 1-yr lagged SOI values ((*c*) mid/late-lactation 2016 (*N* = 5) and (*d*) mid/late-lactation 2009 (*N* = 12)).

material, table S3). The optimal local-scale model explaining the total distance travelled included current year SST and 1-yr lagged chl-*a* (electronic supplementary material, table S3). The total distance travelled was negatively correlated to the current year winter SST (Est: −0.42, LB: −0.77, UB: −0.07), declining by *ca* 180 km across the range of SST observed (figure 3 and table 2). Additionally, 1-yr lagged spring chl-*a* was positively associated with total distance travelled (Est: 3.12, LB: 1.06, UB: 5.21). The total distance travelled by females increased by *ca* 200 km over the observed chl-*a* range (figure 3 and table 2).

## 3.2. Influence of large-scale climate indices on spatial distribution and habitat use

Large-scale climate conditions varied interannually and non-cyclically (electronic supplementary material, table S2). The annual IOD value was lowest in 2005 (−0.04) and highest in 1997 (0.67), with a median value of 0.22. The median SAM value was 0.50, with a minimum in 2002 (−0.51) and a maximum in 2015 (1.61). The SOI value was lowest in 1997 (−11.67) and highest in 2011 (13.30), with a median value of −0.77.

The optimal large-scale model explaining the variation in the 95% home range size estimate for female AUFS included only 1-yr lagged SOI (electronic supplementary material, table S4). Home

**Table 1.** Summary of the annual foraging metrics of female AUFS instrumented between 2001 and 2019 from the Kanowna Island breeding colony, south-eastern Australia. No individuals were deployed with biologging devices in 2004–2005. Values presented are the median [interquartile range] for all metrics except bearing. Bearing is presented as the circular mean ± s.d.

| year | individuals | (trips) | 95% home range size (km$^2$) | trip duration (h) | total distance travelled (km) | maximum range (km) | bearing (°) |
|---|---|---|---|---|---|---|---|
| 2001 | 4 | 27 | 3091.6 [7237.4] | 122.5 [67.6] | 415.8 [150.1] | 143.5 [224.6] | 137.1 ± 75.0 |
| 2002 | 5 | 27 | 2943.1 [2716.1] | 158.4 [129.0] | 519.1 [276.2] | 123.8 [70.9] | 177.7 ± 65.5 |
| 2003 | 9 | 34 | 943.0 [2991.5] | 88.4 [109.1] | 268.1 [332.8] | 65.6 [109.9] | 197.0 ± 75.4 |
| 2004 | 0 | 0 | — | — | — | — | — |
| 2005 | 0 | 0 | — | — | — | — | — |
| 2006 | 2 | 7 | 1338.2 [1153.9] | 124.5 [69.7] | 493.2 [241.9] | 119.1 [36.6] | 198.7 ± 12.0 |
| 2007 | 3 | 20 | 1857.0 [2727.0] | 85.7 [59.5] | 286.7 [175.1] | 74.9 [47.9] | 219.2 ± 65.3 |
| 2008 | 12 | 33 | 1450.4 [3640.0] | 91.7 [110.4] | 221.5 [310.7] | 80.9 [67.5] | 207.4 ± 54.4 |
| 2009 | 12 | 17 | 3151.5 [3860.2] | 182.4 [146.8] | 509.9 [466.0] | 132.2 [102.3] | 167.5 ± 75.8 |
| 2010 | 4 | 6 | 551.8 [3365.5] | 79.8 [128.9] | 158.2 [155.9] | 47.2 [8.0] | 153.3 ± 22.9 |
| 2011 | 8 | 20 | 2867.6 [3559.1] | 132.3 [145.3] | 397.1 [501.6] | 117.4 [109.9] | 164.1 ± 44.4 |
| 2012 | 14 | 26 | 2211.3 [2876.4] | 181.1 [261.4] | 435.8 [502.9] | 117.4 [131.2] | 182.6 ± 52.1 |
| 2013 | 7 | 41 | 791.0 [1723.0] | 90.7 [89.1] | 194.9 [260.5] | 45.4 [65.1] | 191.2 ± 51.8 |
| 2014 | 4 | 24 | 1834.6 [3939.3] | 126.2 [90.1] | 378.6 [153.6] | 96.0 [38.3] | 177.5 ± 39.5 |
| 2015 | 2 | 4 | 3281.7 [1868.3] | 181.9 [90.3] | 501.2 [196.7] | 105.1 [6.5] | 180.5 ± 24.1 |
| 2016 | 3 | 9 | 1472.0 [1336.3] | 104.0 [97.4] | 247.5 [267.4] | 70.5 [47.8] | 191.1 ± 18.1 |
| 2017 | 8 | 29 | 1997.5 [3826.2] | 141.7 [98.0] | 367.5 [237.1] | 86.2 [70.7] | 164.6 ± 55.8 |
| 2018 | 6 | 18 | 2674.0 [2766.7] | 119.8 [31.0] | 361.2 [128.3] | 98.7 [26.4] | 217.6 ± 31.6 |
| 2019 | 7 | 73 | 844.3 [1378.0] | 96.6 [76.1] | 265.5 [197.9] | 57.5 [29.3] | 229.1 ± 43.6 |

range size was positively correlated with 1-yr lagged SOI (Est: 0.03, LB: 0.00, UB: 0.06), with an increase in home range size from *ca* 1800 km$^2$, when 1-yr lagged SOI values were −10, to *ca* 3500 km$^2$, when 1-yr lagged SOI was 10 (figure 4 and table 3).

The optimal large-scale model explaining the total distance travelled included 1-yr lagged IOD and 1-yr lagged SOI (electronic supplementary material, table S4). There was a positive relationship between the total distance travelled and 1-yr lagged SOI (Est: 0.03, LB: 0.02, UB: 0.05), while total distance travelled was negatively correlated with 1-yr lagged IOD (Est: −0.97, LB: −1.69, UB: −0.27). Total distance travelled increased from *ca* 250 km, when 1-yr lagged SOI values were −10, to *ca* 470 km, for 1-yr lagged SOI values of 10 (figures 2 and 3; table 3). Contrastingly, total distance travelled decreased from *ca* 470 km, at 1-yr lagged IOD values of 0, to *ca* 280 km, when 1-yr lagged IOD values were 0.5 (figures 2 and 3; table 3).

Finally, the optimal large-scale model explaining the variation in mean bearing included 1-yr lagged IOD and 1-yr lagged SOI (electronic supplementary material, table S4). The 95% confidence interval of the influence of 1-yr lagged IOD on mean bearing included the null estimate (Est.: 0.13, LB: −0.01, UB: 0.27), and, thus, is considered uncorrelated. By contrast, there was a credible influence of 1-yr lagged SOI on mean bearing (Est.: −0.24, LB: −0.44, UB: −0.5; table 4). Under more positive 1-yr lagged SOI, a greater proportion of individuals had a mean bearing to the north-east of the breeding colony (figures 2 and 5). Neutral 1-yr lagged SOI values were associated with travel/foraging predominantly between south and west of the colony (i.e. central Bass Strait) (figures 2 and 5), while under more negative 1-yr lagged values, the mean bearing was predominantly south-south-west of the colony (figures 2 and 5).

The above analyses were also executed with reduced datasets (i.e. exclusion of years with less than 4 individuals) to explore whether the low-sample-size years unduly influenced the results, but the outcomes were consistent with those reported above.

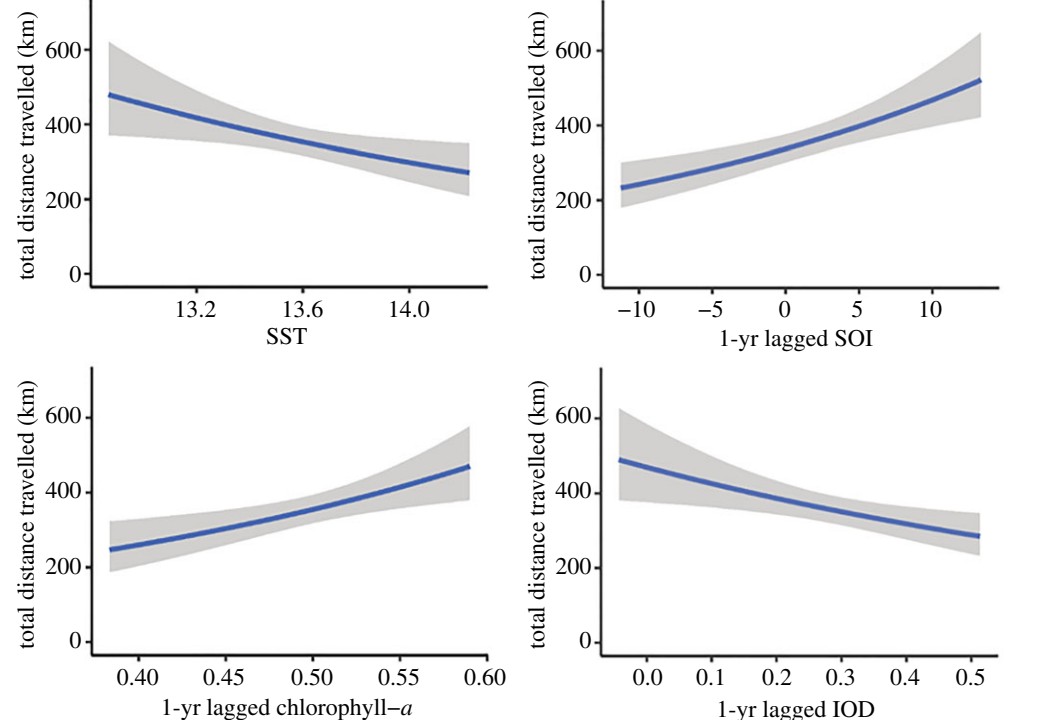

**Figure 3.** Conditional effects plot for the influence of local- and large-scale environmental conditions on the total distance travelled (km) by female AUFS.

**Table 2.** Model estimates for the most supported local-scale model for the total distance travelled by female AUFS instrumented between 2001 and 2019. Est. = estimate. Est. Error = estimate error. LB = lower bound of the 95% credibility interval. UB = upper bound of the 95% credibility interval. The s.d. (intercept) represents the group-level (i.e. ID) effects and shape represents the family-specific parameters.

| response | Cov. | Est. | Est. Error | LB | UB |
|---|---|---|---|---|---|
| 95% home range size (km$^2$) | s.d. (intercept) | 0.70 | 0.09 | 0.54 | 0.88 |
| | intercept | −0.17 | 5.76 | −11.32 | 11.24 |
| | chl-$a_{\text{1-yr lag}}$ | 2.83 | 1.84 | −0.76 | 6.51 |
| | SSHa$_{\text{1-yr lag}}$ | −3.69 | 4.73 | −13.00 | 5.62 |
| | SST$_{\text{1-yr lag}}$ | 0.49 | 0.40 | −0.31 | 1.27 |
| | wind-$u_{\text{1-yr lag}}$ | 0.05 | 0.12 | −0.20 | 0.30 |
| | shape | 0.97 | 0.07 | 0.84 | 1.10 |
| total distance travelled (km) | s.d. (intercept) | 0.38 | 0.06 | 0.27 | 0.50 |
| | intercept | 10.03 | 2.35 | 5.44 | 14.69 |
| | chl-$a_{\text{1-yr lag}}$ | 3.12 | 1.05 | 1.06 | 5.21 |
| | SST | −0.42 | 0.18 | −0.77 | −0.07 |
| | shape | 2.69 | 0.20 | 2.30 | 3.10 |

# 4. Discussion

Previous studies have shown that, like in pelagic foraging marine predators [26], the hunting behaviour and diet of benthic foraging AUFS are influenced by environmental variability [24,31,35]. The results of the present study demonstrate that such environmental variability, particularly with respect to ENSO events, also impacts the spatial distribution and habitat use of female AUFS. Local- and large-scale environmental conditions were associated with changes in the home range size, total distance travelled and the location of foraging activity in female AUFS over a 20-year period.

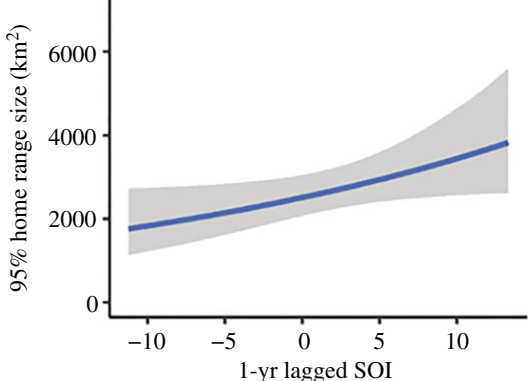

**Figure 4.** Conditional effects plot for the influence of 1-yr lagged SOI on the 95% home range estimate (km$^2$) used by female AUFS.

**Table 3.** Model estimates for the most supported large-scale models for the 95% home range size and total distance travelled by female AUFS instrumented between 2001 and 2019. Est. = estimate. Est. Error = estimate error. LB = lower bound of the 95% credibility interval. UB = upper bound of the 95% credibility interval. The s.d. (intercept) represents the group-level (i.e. ID) effects and shape represents the family-specific parameters.

| response | Cov. | Est. | Est. Error | LB | UB |
|---|---|---|---|---|---|
| 95% home range size (km$^2$) | s.d. (intercept) | 0.68 | 0.08 | 0.53 | 0.85 |
| | intercept | 7.83 | 0.10 | 7.64 | 8.02 |
| | $SOI_{1\text{-yr lag}}$ | 0.03 | 0.02 | 0.00 | 0.06 |
| | shape | 0.97 | 0.07 | 0.85 | 1.10 |
| total distance travelled (km) | s.d. (intercept) | 0.36 | 0.06 | 0.26 | 0.48 |
| | intercept | 6.10 | 0.11 | 5.88 | 6.31 |
| | $SOI_{1\text{-yr lag}}$ | 0.03 | 0.01 | 0.02 | 0.05 |
| | $IOD_{1\text{-yr lag}}$ | −0.97 | 0.36 | −1.69 | −0.27 |
| | shape | 2.68 | 0.20 | 2.30 | 3.09 |

**Table 4.** Model estimates for the most supported model for the mean bearing travelled by female AUFS from the Kanowna Island breeding colony. Est. = estimate. s.d. = standard deviation. LB = lower bound of the 95% credibility interval. UB = upper bound of the 95% credibility interval.

| response | Cov. | Est. | s.d. | LB | UB |
|---|---|---|---|---|---|
| mean bearing from colony (°) | intercept | −2.87 | 0.14 | −3.16 | −2.60 |
| | kappa | 1.01 | 0.19 | 0.65 | 1.39 |
| | $IOD_{1\text{-yr lag}}$ | 0.13 | 0.07 | −0.01 | 0.27 |
| | $SOI_{1\text{-yr lag}}$ | −0.24 | 0.10 | −0.44 | −0.05 |

## 4.1. Influence of environmental variability on habitat use

Understanding the influence of environmental variation on the habitat use of CPF predators is crucial for elucidating environmental impacts on reproductive success and, ultimately, population trajectories. When prey resources shift, provisioning CPF predators may need to increase the duration or distance of foraging trips in order to acquire adequate energy [68]. However, if prey resources shift beyond a critical point, individuals may be unable to obtain enough energy for their own maintenance and to provision their offspring [11,12]. Furthermore, species with concurrent lactation and gestation periods, such as AUFS, need to obtain enough energy to maintain themselves, their growing fetus and their dependant pup [69]. Inadequate prey resources can, thus, result in spontaneous abortion, offspring starvation and/or offspring abandonment [70].

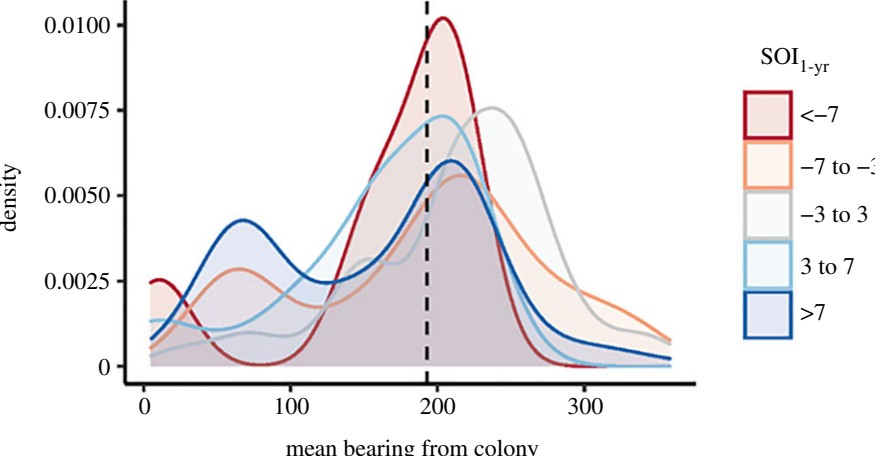

**Figure 5.** Influence of 1-yr lagged SOI on the mean bearing travelled by female AUFS during foraging trips.

While the spatial distribution and habitat use of female AUFS in the present study were influenced by both local- and large-scale environmental variables, much of the variation observed was a result of changes in 1-yr lagged SOI conditions. Individuals travelled a greater distance over a larger area and were more dispersed in their directionality when SOI indices were positive (i.e. towards La Niña) in the previous year. By contrast, under more negative SOI indices (i.e. towards El Niño) in the previous year, individuals remained closer to the colony and were more directed in their travel, travelling predominantly south-southwest of the breeding colony. These results suggest that individuals had a greater foraging effort following years with more positive SOI values than under neutral or negative 1-yr lagged SOI values. This is consistent with previous studies reporting increased foraging effort by female AUFS under more positive 1-yr lagged SOI [24].

ENSO events have been associated with changes in foraging behaviour [25,71], body condition [72] and pup production/survival [73,74] in a range of pinniped species, particularly pelagic foraging pinnipeds. Despite the expectation that benthic species should be more resilient to environmental change [75], several studies report increased pup mortality and higher rates of abortion for benthic foraging pinnipeds during ENSO events. Furthermore, benthic foragers are expected to show greater consistency in foraging behaviour than pelagic foragers due to the predictable distribution of benthic prey [75]. However, higher levels of variability in foraging behaviour were reported in the benthic foraging AUFS than were reported for the pelagic foraging California sealion (*Zalophus californianus*) [76]. Hence, it is reasonable to assume that benthic foragers may also be vulnerable to shifting prey distributions associated with environmental change.

While several studies have reported negative impacts of El Niño events on marine predators (e.g. [25,59,77,78]), these studies have predominantly focused on the impacts of current year, not lagged conditions. It is plausible that there are contrasting effects on marine ecosystems between current El Niño events and years following El Niño events. Additionally, many of the studies investigating ENSO have been in regions of relatively high marine primary productivity, including along the west coast of the Americas. However, the Bass Strait region used by female AUFS is considered to have low marine primary productivity and nutrient flow [41]. This contrast in productivity may explain some of the discrepancy in responses between female AUFS and other marine predators under El Niño or El Niño-like conditions elsewhere. In addition, the influence of ENSO in south-eastern Australia is opposite to that of the Americas, resulting in enhanced upwelling, rather than downwelling, during winter El Niño events [4]. Furthermore, the mechanism driving the impact of El Niño on pelagic predators in the eastern Pacific is largely due to deepening of the thermocline [77], making it harder for air-breathing predators to reach the prey layer, while Bass Strait is shallow and benthic foragers can cover the full depth of the water column.

Female AUFS in the present study also exhibited reduced foraging distances under more positive IOD values in the previous year, suggesting better foraging conditions follow years with more positive IOD values. This seems plausible since the IOD is typically most influential in the austral spring months (September–October) when many of the species consumed by AUFS undergo spawning and larval growth [79–82]. Positive IOD conditions are associated with weakening zonal winds and increasing SSTs in southern Australia. While the increasing temperatures may result in reduced marine primary

productivity, many fish species experience increased growth rates in warmer waters [5]. Thus, positive IOD conditions (and negative SOI conditions, which are also associated with warmer SST) may lead to a greater quantity and/or quality of prey items within Bass Strait in the following year.

Indeed, Kirkwood et al. [31] found that warmer years had increased proportions of red cod (Pseudophycis bachus), barracouta (Thyristes atun) and leatherjackets (Family Triglidae), as well as larger Gould's squid (Nototodarus gouldi), in the AUFS diet, while the proportion of schooling redbait (Emmelichthyis nitidus) in their diet declined. Red cod, leatherjackets and Gould's squid are typically found in benthic habitats and make up a considerable proportion of the AUFS diet [29–31,83,84]. Hence, the replacement of benthic prey species with pelagic redbait in the AUFS diet during cooler years could indicate lower availability of benthic species and presumably worse foraging conditions for female AUFS. Correspondingly, the reduced travel distance and spatial distribution observed under warmer conditions in the current year and the year following more negative SOI and/or positive IOD suggest that individuals were able to meet the energetic demands of pup provisioning and individual maintenance with greater ease. The increased benthic foraging efficiency in response to more positive 2-yr lagged IOD reported for the study species [24] provides further support for such an increase in prey availability or quality. The 1-year later AUFS responses also allow future development of forecasts based on real-time climate indices.

Alternatively, the observed changes in spatial distribution may be associated with cold water inflow into Bass Strait. The inflow of cold, nutrient-rich SASW from the southwest can improve the local marine productivity [41], creating better foraging conditions for marine predators. However, the variability in chl-a concentrations observed in the present study was quite minimal (varying by less than $0.1 \, \mathrm{mg \, m^{-3}}$ between most years) and the increased foraging effort following years with higher chl-a does not support an increase in prey availability associated with SASW intrusion. Furthermore, this seasonal intrusion into Bass Strait, occurring in the austral winter [85], has not been found to be influenced by SOI or IOD.

ENSO conditions have been associated with regional upwelling activity in south-eastern Australia [86], with upwelling activity strengthened and weakened during El Niño and La Niña events, respectively [4]. Interestingly, the increase in total distance travelled with increasing chl-a concentrations in the previous year suggests that upwelling activity may not be driving the influence of 1-yr lagged SOI observed, which may be expected given the El Niño-enhanced upwelling known to occur in the region. There is little nutrient transport into the central Bass Strait [85] from the west and east shelf-slopes, where much of the upwelling activity occurs [86], suggesting more localized influences on marine primary productivity within the AUFS foraging range. Nonetheless, the observed increase in foraging effort following years with higher chl-a is consistent with previous studies of the species which suggest this may be driving changes in the availability or profitability of AUFS prey, though the mechanisms behind this are unclear [35]. Increased marine primary productivity is known to improve pelagic prey availability and may result in greater proportions of pelagic baitfish within Bass Strait. While similar in nutritional content [87], pelagic baitfish are more difficult for female AUFS to capture compared with benthic prey [87], leading to lower profitability and, hence, greater foraging effort and altered habitat use in female AUFS under high chl-a conditions. However, if there is an abundance of pelagic baitfish but few benthic prey available, it may be more efficient to capture pelagic prey. Hence, the payoff for foraging on baitfish could alter depending on their abundance [88].

## 4.2. Implications for benthic central-place foragers under future environmental change

With the frequency and severity of extreme climate events associated with climate drivers expected to continue increasing [15,16,89,90], it is important to identify how CPF predators respond to environmental change. Long-term studies provide an opportunity to investigate relationships between habitat use and the conditions experienced in the marine environment. While several studies have highlighted the susceptibility of pelagic predators to environmental perturbations, few studies have reported impacts on CPF benthic predators.

Marine ecosystems as a whole have experienced significant negative effects of environmental change [89,91], including shifts in species' range, community structure, phenology, recruitment, growth and reproductive success (e.g. [6,23,59,92–95]). These impacts are predicted to continue under anticipated climate change due to the interplay between climate drivers, such as ENSO, and local environmental conditions [89,90]. Consequently, CPF species will need to exhibit plasticity in not only their diving behaviour and diet but also their habitat use in response to the environmental extremes experienced at short (i.e. intra-annual) and long (i.e. inter-decadal) time scales. This puts CPF species at risk due

to the foraging constraints on provisioning adults that need to return to land at regular intervals to provision offspring. While it is assumed that this applies predominantly to pelagic predators, the results of the present study indicate that benthic CPF species are also at risk under future environmental change, even within a region of comparatively low marine primary productivity.

In summary, the present study provided empirical evidence of altered habitat use in the benthic foraging AUFS in response to oceanographic change at both the local and large scales. Shifts in habitat use and spatial distribution may have significant consequences for CPF predators, including lower body condition [96] and, ultimately, reduced reproductive success [13,97]. The results suggest that climate-mediated changes in prey availability, distribution and/or quality are driving the changes observed. Anticipated climate change is likely to have significant consequences for marine environments, including benthic habitats, through changes in marine primary productivity and prey availability.

Ethics. All work was carried out with the approval of the Deakin University Animal Ethics committee and under Department of Environment, Land, Water and Planning (previously Department of Sustainability and Environment, Victoria, Australia) wildlife research permits (10000187, 10000706, 10001143, 10001672, 10005362 and 10005848).

Data accessibility. Data used in the analyses are provided in the electronic supplementary material. Code used in the analyses is available via Github: https://github.com/CSpeakman/Speakman_AUFS_space_use and have been archived within the Zenodo repository: https://zenodo.org/badge/latestdoi/392555183.

The data are provided in electronic supplementary material [98].

Authors' contributions. C.N.S. and J.P.Y.A. conceived and designed the experiments. C.N.S. and J.P.Y.A. performed the experiments. C.N.S., A.J.H., A.J.H., J.R.H. and J.P.Y.A. acquired the data. C.N.S. and A.J.H. analysed the data. All authors contributed to manuscript preparation and gave final approval for publication.

Competing interests. We declare we have no competing interests

Funding. This work was supported by the Australian Research Council (grant nos. DP0664167 and DP110102065); Office of Naval Research Marine Mammals and Biological Oceanography Program Award (grant no. N00014-10-1-0385); Sea World Research and Rescue Foundation (grant no. SWR42020); Winnifred Violet Scott Trust (grant no. WVSE/08-11); and Holsworth Wildlife Research Endowment (grant no. 06153).

Acknowledgements. We thank the many researchers, students and volunteers who have assisted in the data collection during the study period. Logistical support was provided by Parks Victoria, Prom Adventurer Charters and Best Helicopters, and the assistance of the many Parks Victoria rangers involved, Geoff Boyd, and Sean Best is gratefully acknowledged.

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
