## [Peer Review File · Royal Society Open Science]

Review History

RSOS-211052.R0 (Original submission)

Review form: Reviewer 1

Is the manuscript scientifically sound in its present form?

Yes

Are the interpretations and conclusions justified by the results?

Yes

Is the language acceptable?

Yes

Do you have any ethical concerns with this paper?

No

Have you any concerns about statistical analyses in this paper?

No

Recommendation?

Accept with minor revision (please list in comments)

Comments to the Author(s)

I enjoyed reading the manuscript, it is scientifically sound, well written and to the point, and advances knowledge on the potential impacts of climatic changes on the benthically foraging AUFS. It would be nice to see the relationships with indices of reproductive performance/offspring growth, if such exist for a similarly long time period. I have the following minor comments which should be addressed:

The sentence on p4 lines 38-43: 'However benthic systems are considered more stable and,lower susceptibility to environmental variability.' does not quite anticipate the second last sentence of the next paragraph: '.... less is known about the temporal variability of benthic habitat communities, and, thus, the influence of environmental change'

I suggest to delete the last sentence of the second paragraph.

Lines15-22 on p6: 'While it is known that environmental variation, it is important to determine how environmental change impacts benthic marine communities.' The last bit of this sentence after the colony also seems a bit out of place. This study does not determine how environmental change impacts benthic communities. This sentence should be rephrased.

Lines 31-36 on p7. I suggest this sentence be rewritten as: 'To aid identification, individually numbered plastic tags (Super Tags, Dalton, Woolgoolga, Australia) were inserted into the trailing edge of each fore flipper before the individual was allowed to recover from anaesthesia and resume normal behaviour.'

Environmental variables on p8 and p9: it's not quite clear to me why data from before the year when deployments began are incorporated in analyses. I.e., after accounting for lags of 1-2 years, why are data before 1999 included? Also in the results on p11, an SST low for 1998 is reported, a few years before the first deployments...and again for IOD and SOI in 1997. Why are the long terms means and variability calculated on data inclusive of years before those used in the model analyses? If I am not understanding something correctly, perhaps better explanation is needed, because at the moment it is a bit confusing.

L12 on p10: 'ran' should be 'run' (?)

A full stop is missing at line 52 of p12.

L33 on P13: 'influenced'

The Discussion becomes a little fuzzy at line 19 of p 16, where the possibility that (alternatively to what has been considered or shown in the models) inflow of SASW may account for the observed spatial changes in behavior. This is introduced suddenly but not discussed further at all. The next sentence, which is the last sentence of this paragraph, starts with the word 'However', but does not actually address the idea above it. Instead it talks about ENSO conditions and upwelling (that are not related to SASW), which are then enlarged upon in the following paragraph. I think that the last sentence of the paragraph starting at line 19, should be the first sentence of the paragraph that follows it (with the first word 'However' removed), and there needs to be more said about the possible SASW influence in the former paragraph, or at least explanation as to why this could not be accounted for in the study.

Also, the sentence starting at the bottom of p16: 'While similar in nutritional content [76], pelagic baitfish are more difficult for female AUFS to capture compared with benthic prey [76], leading to

lower profitability and, hence, greater foraging effort and altered habitat use in female AUFS under high chl-a conditions.' There is greater foraging effort and altered habitat use if the same conditions that are good for baitfish are bad for the normal benthic prey, or the ability to access them. This needs to be made clearer.

Lines 40-42 on P17: 'Consequently, CPF species will need to exhibit plasticity in not only their foraging behavior but also their diet and habitat use'. I would think that the animal's choices on where to forage (habitat use) and what to eat (diet) are aspects of foraging behavior? Therefore I am not sure of why the word 'also' is used.

Review form: Reviewer 2

Is the manuscript scientifically sound in its present form?

Yes

Are the interpretations and conclusions justified by the results?

Yes

Is the language acceptable?

Yes

Do you have any ethical concerns with this paper?

No

Have you any concerns about statistical analyses in this paper?

No

Recommendation?

Major revision is needed (please make suggestions in comments)

Comments to the Author(s)

Influence of environmental variation on spatial distribution and habitat-use in a benthic foraging marine predator

This is an interesting and relevant paper from an extensive long-term dataset making this a very important paper for describing potential effects of climate change on benthic foragers in the marine environment. Overall, this is a well written manuscript from a grammar point of view and I have very few specific comments which are all listed below. However, I do have a few larger subjects that need addressed including: how the data is presented; the lack of referral to the fact that for 2 years you have no female data and for several other years very low data and how this may influences results and conclusions; in general how you know your sample size in any year is representative of AFSFs; expansion of the discussion including discussion of these results in terms of what has been found for other benthic Otariids and how these results differ in magnitude to the results found for pelagic species (which you repeatedly say have been looked at in this way – so compare them please).

Major Issues

Methods

I have no problem with the statistical analysis however I would like the paper to be more descriptive showing the biological data behind the statistics - ie Tabel S1 shows the summary

statistics for each female, I would like to see a table within the paper showing a summary of this data specifically how many females were captured each year and summarising each years home ranges, distances etc in that table, in the manuscript – not as additional data.

There are two years (2004 and 2005) when no data for any females are shown which is in complete contradiction to what is stated in the methods which is “Sampling occurred during the austral winter (April-August), the period of peak nutritional demand for lactating females each year between 2001-2019.” This is not mentioned in the paper. Additionally, there are very low numbers of females (≤ 3) in 2006, 2007, 2015 and 2016. These 6 years are some of the years where some of the minimum or maximum IOD, SAM, and SOI's were recorded. I think you need to check if this makes an influence on your results particularly for the 4 years 2004 to 2007 as this is an extended time period for your 1 & 2 yr lag model with little or no actual female foraging information. If you run the analysis without these years, or only after these years does it make a difference, if no, then that can be reported, if yes, than it needs explaining. I do not think this makes the analysis or paper any less relevant or important it is just not being made clear in the paper and how this may affect the results. This needs undertaking and explaining.

Results

In general although your sample size extends over many years which is phenomenal in today's world, your sample size compared to colony size is small, how do you know these females are representative. This needs explaining including references to other studies.

Discussion

I find the discussion is a bit repetitive of the introduction and of itself. I would like to see it broader given the topic of this manuscript. I would like to see a short discussion putting these results in context with both: a) other benthic Otariids (predominantly Galapagos / Australian / South American sea lions) particularly as they are all affected by SOI and; b) How difference and how big a shift is this relative to results already known for pelagic species? Given you say in the introduction and discussion that for pelagic species these effects has been documented. There is no discussion of either in here and they are needed.

Figures and Tables

I think the current Tables 1 & 2 could go in supplementary data and that a biological data summary table needs to go in the paper i.e. number of females in each year and a mean of those years KHR, distance etc. And for Table S1 and its summary in the manuscript there the animals need to be listed in order by year – currently the first six or so years data are all jumbled by year.

Minor changes

Introduction Ln 26 please define foraging effort in the context you are using here – you use it through out to refer to home range used and time/distance from shore – however there are other description of foraging effort so you need to clearly define what you are calling foraging effort for this paper

Pg 4 Ln 13 I think there should be an “as the” or “because the” in this sentence ie This is particularly concerning BECAUSE THE balance of phases of large-scale climate drives are predicted to change over the coming decades”

Methods

How were the females chosen at random? 3400 pups and therefor females is a very large colony – were the females always from one end of the colony? From the area of the colony most accessible? Evenly or randomly spaced throughout the colony?

Pg 14 Ln 12 Please add a reference relating to the sentence that ends "...dependant pup."

Pg 18 sentence starting "Shifts in habitat use" please add a reference after lower body condition and at the end of the sentence

General for discussion

For Bass Straight, there is extensive commercial fishing allowed, particularly bottom trawling. Influences and changes of fishing intensity over this time just needs a mention as over the last 20 years I would be betting that fishing effort, methods., allowed catch etc may deplete prey at differing times but can also alter the benthos - and it may be worth mentioning as a variable that was not looked at but should be considered for a complete overall picture.

Figure 1 Please make the colony blue circle - black it is very hard to pick out at the moment and impossible if the paper is printed in black and white (which is what I did)

Decision letter (RSOS-211052.R0)

Dear Ms Speakman

The Editors assigned to your paper RSOS-211052 "Influence of environmental variation on spatial distribution and habitat-use in a benthic foraging marine predator" have now received comments from reviewers and would like you to revise the paper in accordance with the reviewer comments and any comments from the Editors. Please note this decision does not guarantee eventual acceptance.

Please submit your revised manuscript and required files (see below) no later than 21 days from today's (ie 29-Jul-2021) date. Note: the ScholarOne system will 'lock' if submission of the revision is attempted 21 or more days after the deadline. If you do not think you will be able to meet this deadline please contact the editorial office immediately.

on behalf of Dr Asha de Vos (Associate Editor) and Pete Smith (Subject Editor)
openscience@royalsociety.org

Associate Editor Comments to Author (Dr Asha de Vos):

Associate Editor: 1

Comments to the Author:

This paper has been well received and thoroughly reviewed. I don't have more to add but with the edits as recommended, this paper will be a valuable addition.

Associate Editor: 2

Comments to the Author:

(There are no comments.)

Reviewer comments to Author:

Reviewer: 1

Comments to the Author(s)

I enjoyed reading the manuscript, it is scientifically sound, well written and to the point, and advances knowledge on the potential impacts of climatic changes on the benthically foraging AUFS. It would be nice to see the relationships with indices of reproductive performance/offspring growth, if such exist for a similarly long time period. I have the following minor comments which should be addressed:

The sentence on p4 lines 38-43: 'However benthic systems are considered more stable and,lower susceptibility to environmental variability.' does not quite anticipate the second last sentence of the next paragraph: '.... less is known about the temporal variability of benthic habitat communities, and, thus, the influence of environmental change'

I suggest to delete the last sentence of the second paragraph.

Lines15-22 on p6: 'While it is known that environmental variation, it is important to determine how environmental change impacts benthic marine communities.' The last bit of this sentence after the colony also seems a bit out of place. This study does not determine how environmental change impacts benthic communities. This sentence should be rephrased.

Lines 31-36 on p7. I suggest this sentence be rewritten as: 'To aid identification, individually numbered plastic tags (Super Tags, Dalton, Woolgoolga, Australia) were inserted into the trailing edge of each fore flipper before the individual was allowed to recover from anaesthesia and resume normal behaviour.'

Environmental variables on p8 and p9: it's not quite clear to me why data from before the year when deployments began are incorporated in analyses. I.e., after accounting for lags of 1-2 years, why are data before 1999 included? Also in the results on p11, an SST low for 1998 is reported, a few years before the first deployments...and again for IOD and SOI in 1997. Why are the long terms means and variability calculated on data inclusive of years before those used in the model

analyses? If I am not understanding something correctly, perhaps better explanation is needed, because at the moment it is a bit confusing.

L12 on p10: 'ran' should be 'run' (?)

A full stop is missing at line 52 of p12.

L33 on P13: 'influenced'

The Discussion becomes a little fuzzy at line 19 of p 16, where the possibility that (alternatively to what has been considered or shown in the models) inflow of SASW may account for the observed spatial changes in behavior. This is introduced suddenly but not discussed further at all. The next sentence, which is the last sentence of this paragraph, starts with the word 'However', but does not actually address the idea above it. Instead it talks about ENSO conditions and upwelling (that are not related to SASW), which are then enlarged upon in the following paragraph. I think that the last sentence of the paragraph starting at line 19, should be the first sentence of the paragraph that follows it (with the first word 'However' removed), and there needs to be more said about the possible SASW influence in the former paragraph, or at least explanation as to why this could not be accounted for in the study.

Also, the sentence starting at the bottom of p16: 'While similar in nutritional content [76], pelagic baitfish are more difficult for female AUFS to capture compared with benthic prey [76], leading to lower profitability and, hence, greater foraging effort and altered habitat use in female AUFS under high chl-a conditions.' There is greater foraging effort and altered habitat use if the same conditions that are good for baitfish are bad for the normal benthic prey, or the ability to access them. This needs to be made clearer.

Lines 40-42 on P17: 'Consequently, CPF species will need to exhibit plasticity in not only their foraging behavior but also their diet and habitat use'. I would think that the animal's choices on where to forage (habitat use) and what to eat (diet) are aspects of foraging behavior? Therefore I am not sure of why the word 'also' is used.

Reviewer: 2

Comments to the Author(s)

Influence of environmental variation on spatial distribution and habitat-use in a benthic foraging marine predator

This is an interesting and relevant paper from an extensive long-term dataset making this a very important paper for describing potential effects of climate change on benthic foragers in the marine environment. Overall, this is a well written manuscript from a grammar point of view and I have very few specific comments which are all listed below. However, I do have a few larger subjects that need addressed including: how the data is presented; the lack of referral to the fact that for 2 years you have no female data and for several other years very low data and how this may influences results and conclusions; in general how you know your sample size in any year is representative of AFSFs; expansion of the discussion including discussion of these results in terms of what has been found for other benthic Otariids and how these results differ in magnitude to the results found for pelagic species (which you repeatedly say have been looked at in this way – so compare them please).

Major Issues

Methods

I have no problem with the statistical analysis however I would like the paper to be more descriptive showing the biological data behind the statistics - ie Tabel S1 shows the summary statistics for each female, I would like to see a table within the paper showing a summary of this data specifically how many females were captured each year and summarising each years home ranges, distances etc in that table, in the manuscript - not as additional data.

There are two years (2004 and 2005) when no data for any females are shown which is in complete contradiction to what is stated in the methods which is "Sampling occurred during the austral winter (April-August), the period of peak nutritional demand for lactating females each year between 2001-2019." This is not mentioned in the paper. Additionally, there are very low numbers of females (≤ 3) in 2006, 2007, 2015 and 2016. These 6 years are some of the years where some of the minimum or maximum IOD, SAM, and SOI's were recorded. I think you need to check if this makes an influence on your results particularly for the 4 years 2004 to 2007 as this is an extended time period for your 1 & 2 yr lag model with little or no actual female foraging information. If you run the analysis without these years, or only after these years does it make a difference, if no, then that can be reported, if yes, than it needs explaining. I do not think this makes the analysis or paper any less relevant or important it is just not being made clear in the paper and how this may affect the results. This needs undertaking and explaining.

Results

In general although your sample size extends over many years which is phenomenal in todays world, your sample size compared to colony size is small, how do you know these females are representative. This needs explaining including references to other studies.

Discussion

I find the discussion is a bit repetitive of the introduction and of itself. I would like to see it broader given the topic of this manuscript. I would like to see a short discussion putting these results in context with both: a) other benthic Otariids (predominantly Galapogas / Australian / South American sea lions) particularly as they are all affected by SOI and; b) How difference and how big a shift is this relative to results already known for pelagic species? Given you say in the introduction and discussion that for pelagic species these effects has been documented. There is no discussion of either in here and they are needed.

Figures and Tables

I think the current Tables 1 & 2 could go in supplementary data and that a biological data summary table needs to go in the paper i.e. number of females in each year and a mean of those years KHR, distance etc. And for Table S1 and its summary in the manuscript there the animals need to be listed in order by year - currently the first six or so years data are all jumbled by year.

Minor changes

Introduction Ln 26 please define foraging effort in the context you are using here - you use it through out to refer to home range used and time/distance from shore - however there are other description of foraging effort so you need to clearly define what you are calling foraging effort for this paper

Pg 4 Ln 13 I think there should be an "as the" or "because the" in this sentence ie This is particularly concerning BECAUSE THE balance of phases of large-scale climate drives are predicted to change over the coming decades"

Methods

How were the females chosen at random? 3400 pups and therefore females is a very large colony – were the females always from one end of the colony? From the area of the colony most accessible? Evenly or randomly spaced throughout the colony?

Pg 14 Ln 12 Please add a reference relating to the sentence that ends “...dependant pup.”

Pg 18 sentence starting “Shifts in habitat use” please add a reference after lower body condition and at the end of the sentence

General for discussion

For Bass Straight, there is extensive commercial fishing allowed, particularly bottom trawling. Influences and changes of fishing intensity over this time just needs a mention as over the last 20 years I would be betting that fishing effort, methods., allowed catch etc may deplete prey at differing times but can also alter the benthos – and it may be worth mentioning as a variable that was not looked at but should be considered for a complete overall picture.

Figure 1 Please make the colony blue circle – black it is very hard to pick out at the moment and impossible if the paper is printed in black and white (which is what I did)

===PREPARING YOUR MANUSCRIPT===

===PREPARING YOUR REVISION IN SCHOLARONE===

Author's Response to Decision Letter for (RSOS-211052.R0)

See Appendix A.

RSOS-211052.R1 (Revision)

Review form: Reviewer 1

Is the manuscript scientifically sound in its present form?

Yes

Are the interpretations and conclusions justified by the results?

Yes

Is the language acceptable?

Yes

Do you have any ethical concerns with this paper?

No

Have you any concerns about statistical analyses in this paper?

No

Recommendation?

Accept as is

Comments to the Author(s)

Reviewer comments have been adequately addressed.

Review form: Reviewer 2

Is the manuscript scientifically sound in its present form?

Yes

Are the interpretations and conclusions justified by the results?

Yes

Is the language acceptable?

Yes

Do you have any ethical concerns with this paper?

No

Have you any concerns about statistical analyses in this paper?

No

Recommendation?

Accept as is

Comments to the Author(s)

Well done the paper is clearer and your revision has covered my concerns with the data presentation and modelling

Decision letter (RSOS-211052.R1)

Dear Ms Speakman,

It is a pleasure to accept your manuscript entitled "Influence of environmental variation on spatial distribution and habitat-use in a benthic foraging marine predator" in its current form for publication in Royal Society Open Science. The comments of the reviewer(s) who reviewed your manuscript are included at the foot of this letter.

Kind regards,

Royal Society Open Science Editorial Office
Royal Society Open Science
openscience@royalsociety.org

on behalf of Dr Asha de Vos (Associate Editor) and Pete Smith (Subject Editor)
openscience@royalsociety.org

Associate Editor Comments to Author (Dr Asha de Vos):

Associate Editor: 1

Comments to the Author:

Thank you for your submission. I am very happy with the reviews - as you would be. Looking forward to the publication.

Associate Editor: 2

Comments to the Author:

Thank you for reverting with the revisions. Given they were 'major' revisions, I am requesting that the manuscript go back for review before a final decision is made.

Reviewer comments to Author:

Reviewer: 2

Comments to the Author(s)

Well done the paper is clearer and your revision has covered my concerns with the data presentation and modelling

Reviewer: 1

Comments to the Author(s)

Reviewer comments have been adequately addressed.

Appendix A

Dear Dr Asha de Vos,

We thank the Associate Editor and reviewers for their detailed and constructive feedback on the manuscript "Influence of environmental variation on spatial distribution and habitat-use in a benthic foraging marine predator". We have integrated these suggestions, where appropriate, into the manuscript and feel this has greatly strengthened and clarified the manuscript.

The main changes to the manuscript include:

- 1) Reviewer 2 suggested that the sample size in some years may be influencing the results of the study. As such, we reanalysed the data to exclude years with <4 individuals. The outputs of the models were consistent with the reported results and, thus, we feel confident in reporting the original results.
- 2) We have attempted to improve the contextualisation within the existing literature, particularly in relation to the influence of El Niño Southern Oscillation events on otariids. This required the addition of a paragraph in the discussion.
- 3) We have clarified how animals were captured while maintaining minimal sampling bias. The '*Animal handling and instrumentation*' section of the Methods was, thus, lengthened and had some rephrasing.
- 4) An additional table was added to the main text summarising the annual foraging metrics and sample size. Additionally, the model comparison tables were moved to the supplementary material as suggested by Reviewer 2.

Detailed responses to the reviewer comments are included below. The manuscript has been supplied with track changes and as a finalised copy without track changes. Please note that the page and line numbers given in the point-by-point response are based on the clean version.

We look forward to hearing the outcome of the next round of reviews.

Warm regards,
Cassie Speakman

Reviewer: 1

I enjoyed reading the manuscript, it is scientifically sound, well written and to the point, and advances knowledge on the potential impacts of climatic changes on the benthically foraging AUFs. It would be nice to see the relationships with indices of reproductive performance/offspring growth, if such exist for a similarly long time period.

While such data does exist, it is outside of the scope of the present study, but it is an avenue of exploration for future studies.

The sentence on **p4 lines 38-43**: 'However benthic systems are considered more stable and,lower susceptibility to environmental variability.' does not quite anticipate the second last sentence of the next paragraph: '.... less is known about the temporal variability of benthic habitat communities, and, thus, the influence of environmental change' I suggest to delete the last sentence of the second paragraph.

While we understand why this was suggested, we believe the two statements provide required context for the study and highlight the lack of information on benthic communities and predators, particularly with respect to environmental change. As such, we have opted to not remove the sentence 'However benthic systems..... to environmental variability' from the manuscript.

Lines15-22 on p6: 'While it is known that environmental variation, it is important to determine how environmental change impacts benthic marine communities.' The last bit of this sentence after the colony also seems a bit out of place. This study does not determine how environmental change impacts benthic communities. This sentence should be rephrased.

Good point. The statement has been rephrased, putting the emphasis on the benthic predators rather than the communities. [L60]

Lines 31-36 on p7. I suggest this sentence be rewritten as: 'To aid identification, individually numbered plastic tags (Super Tags, Dalton, Woolgoolga, Australia) were inserted into the trailing edge of each fore flipper before the individual was allowed to recover from anaesthesia and resume normal behaviour.'

Nice suggestion. Text amended to the recommended rephrasing. [L145-146]

Environmental variables on p8 and p9: it's not quite clear to me why data from before the year when deployments began are incorporated in analyses. I.e., after accounting for lags of 1-2 years, why are data before 1999 included? Also in the results on p11, an SST low for 1998 is reported, a few years before the first deployments...and again for IOD and SOI in 1997. Why are the long term means and variability calculated on data inclusive of years before those used in the model analyses? If I am not understanding something correctly, perhaps better explanation is needed, because at the moment it is a bit confusing.

Sorry about the confusion. Overall, data were extracted for those time periods but only the 2-yr lag was needed (i.e., 1999 onwards). I have amended the text accordingly. [L183 & 185]

L12 on p10: 'ran' should be 'run' (?)

This tends to be written as 'ran' given it is past tense. As such, I have left it as it was.

A full stop is missing at **line 52 of p12**.

Full stop added. [L280]

L33 on P13: 'influenced'

Amended. [L299]

The Discussion becomes a little fuzzy at **line 19 of p 16**, where the possibility that (alternatively to what has been considered or shown in the models) inflow of SASW may account for the observed spatial changes in behavior. This is introduced suddenly but not discussed further at all. The next sentence, which is the last sentence of this paragraph, starts with the word 'However', but does not actually address the idea above it. Instead it talks about ENSO conditions and upwelling (that are not related to SASW), which are then enlarged upon in the following paragraph. I think that the last sentence of the paragraph starting at line 19, should be the first sentence of the paragraph that follows it (with the first word 'However' removed), and there needs to be more said about the possible SASW influence in the former paragraph, or at least explanation as to why this could not be accounted for in the study.

This is a good point. We have moved the last sentence (starting with 'However, ENSO..' with 'However' removed) to the start of the following paragraph as suggested. We have also included further discussion on the SASW. [L382-389]

Also, the sentence starting at the **bottom of p16**: 'While similar in nutritional content [76], pelagic baitfish are more difficult for female AUFS to capture compared with benthic prey [76], leading to lower profitability and, hence, greater foraging effort and altered habitat use in female AUFS under high chl-a conditions.' There is greater foraging effort and altered habitat use if the same conditions that are good for baitfish are bad for the normal benthic prey, or the ability to access them. This needs to be made clearer.

Good point. We have added a caveat in the text to indicate that the pay-off to foraging on baitfish could alter depending on their abundance. [L403-405]

Lines 40-42 on P17: 'Consequently, CPF species will need to exhibit plasticity in not only their foraging behavior but also their diet and habitat use'. I would think that the animal's choices on where to forage (habitat use) and what to eat (diet) are aspects of foraging behavior? Therefore I am not sure of why the word 'also' is used.

Agreed, they are aspects (or outcomes) of foraging behaviour. The distinction for diet, habitat use (spatial distribution) and diving behaviour was the intended distinction, not foraging behaviour per se. I have amended this to clarify. [L419-420]

Reviewer: 2

This is an interesting and relevant paper from an extensive long-term dataset making this a very important paper for describing potential effects of climate change on benthic foragers in the marine environment. Overall, this is a well written manuscript from a grammar point of view and I have very few specific comments which are all listed below. However, I do have a few larger subjects that need addressed including: how the data is presented; the lack of referral to the fact that for 2 years you have no female data and for several other years very low data and how this may influences results and conclusions; in general how you know your sample size in any year is representative of AFSFs; expansion of the discussion including discussion of these results in terms of what has been found for other benthic Otariids and how these results differ in magnitude to the results found for pelagic species (which you repeatedly say have been looked at in this way – so compare them please).

Please see the responses to the above concerns in the detailed comments below.

Major Issues

Methods

I have no problem with the statistical analysis however I would like the paper to be more descriptive showing the biological data behind the statistics - ie Tabel S1 shows the summary statistics for each female, I would like to see a table within the paper showing a summary of this data specifically how many females were captured each year and summarising each years home ranges, distances etc in that table, in the manuscript – not as additional data.

I have included the annual foraging metrics (now Table 1) including the number of females, home ranges, distances, etc. We have kept the summaries for each individual in Table S1 as well.

There are two years (2004 and 2005) when no data for any females are shown which is in complete contradiction to what is stated in the methods which is “Sampling occurred during the austral winter (April-August), the period of peak nutritional demand for lactating females each year between 2001-2019.” This is not mentioned in the paper.

I have amended the text to make it clear that no individuals were instrumented with biologging devices in 2004-2005. [L118]

Additionally, there are very low numbers of females (≤ 3) in 2006, 2007, 2015 and 2016. These 6 years are some of the years where some of the minimum or maximum IOD, SAM, and SOI's were recorded. I think you need to check if this makes an influence on your results particularly for the 4 years 2004 to 2007 as this is an extended time period for your 1 & 2 yr lag model with little or no actual female foraging information. If you run the analysis without these years, or only after these years does it make a difference, if no, then that can be reported, if yes, than it needs explaining. I do not think this makes the analysis or paper any less relevant or important it is just not being made clear in the paper and how this may affect the results. This needs undertaking and explaining.

This is a very valid point. However, the modelling framework used was selected due to the robustness around differing sample sizes and 'gaps' in the data.

Nonetheless, I have reran the analyses for the climate index models as requested. For the reanalyses, the 2004-2007 year period was excluded (i.e., 2006-2007 removed from the dataset). Additionally, I ran the models with 2015-2016 also excluded (i.e., no years had <4 individuals in the analyses).

The outputs are as below. For reference, the optimal models are:

- H₇ for home range size includes 1-yr lagged SOI
- H₇ for distance travelled includes 1-yr lagged SOI and 1-yr lagged IOD.
 - When 2004-2007 are excluded, H₇ also includes 2-yr lagged IOD.
- H₇ for bearing includes 1-yr lagged SOI and 1-yr lagged IOD.

In Table 1 below, I have colour graded the values for easier visualisation of the similarities and differences. As you can see, the models are highly comparable between the original analysis, the 2004-2007 exclusion, and the 2004-2007 and 2015-2016 exclusion. The main difference is that some of the model weightings are a little different, but the optimal model remains the same (apart from H₇ for total distance travelled, which also includes 2-yr lagged IOD).

The model estimates (from each H₇ model) are also almost identical apart from the confidence around the influence of 1-yr lagged IOD on bearing. This became a ‘significant’ result when 2004-2007 and 2015-2016 were excluded from the model (highlighted in the red text in Table 2 below). All other results remained the same.

As such, we feel confident in reporting the findings as is (i.e., including 2004-2007 and 2015-2016).

Table 1

Response	Model	Original	2004-2007 excl.	+2015-2016 excl.	Original	2004-2007 excl.	+2015-2016 excl.	Original	2004-2007 excl.	+2015-2016 excl.
		ΔELPD	ΔELPD	ΔELPD	ΔELPD SE	ΔELPD SE	ΔELPD SE	LOO weight	LOO weight	LOO weight
95% Home range size (km ²)	H ₀	-34.0	-32.3	-27.8	8.0	8.0	7.4	0.03	0.00	0.00
	H ₁	-3.4	-2.8	-3.1	1.2	1.3	1.4	0.00	0.04	0.04
	H ₂	-2.0	-1.6	-2.2	0.9	1.0	1.1	0.00	0.11	0.08
	H ₃	-3.3	-2.5	-2.6	1.9	1.9	2.0	0.00	0.07	0.09
	H ₄	-3.2	-2.8	-2.3	1.1	1.2	1.0	0.00	0.04	0.06
	H ₅	-2.2	-1.1	-0.9	1.0	0.6	0.8	0.00	0.16	0.19
	H ₆	-2.3	-1.5	-2.4	1.7	1.7	1.8	0.00	0.16	0.09
H ₇	0.0	0.0	0.0	0.0	0.0	0.0	0.97	0.42	0.45	
Total distance travelled (km)	H ₀	-14.6	-23.1	-21.7	14.8	12.5	12.2	0.17	0.04	0.05
	H ₁	-1.7	-1.6	-2.2	2.0	2.1	1.5	0.00	0.08	0.06
	H ₂	-1.0	-1.9	-1.3	1.6	2.0	1.0	0.00	0.06	0.10
	H ₃	-5.3	-4.8	-5.1	2.8	2.6	3.1	0.00	0.02	0.02
	H ₄	-0.9	-0.3	-1.0	1.6	1.8	1.1	0.00	0.23	0.15
	H ₅	-0.1	-0.4	-0.4	1.2	1.7	0.4	0.32	0.22	0.23
	H ₆	-4.4	-4.2	-4.4	2.7	2.7	3.2	0.00	0.04	0.05
H ₇	0.0	0.0	0.0	0.0	0.0	0.0	0.51	0.32	0.34	
Response	Model	LPPD	LPPD	LPPD	ΔWAIC	ΔWAIC	ΔWAIC			
Mean bearing from colony (°)	H ₀	-178.82	-170.43	-162.02	3.19	3.10	4.62			
	H ₁	-173.47	-164.83	-155.11	9.32	11.96	11.07			
	H ₂	-173.93	-165.90	-156.51	5.05	5.27	4.38			
	H ₃	-176.23	-167.91	-159.00	10.11	9.34	10.30			
	H ₄	-173.54	-164.82	-155.86	4.11	4.27	5.63			
	H ₅	-174.93	-166.55	-157.72	1.53	1.91	1.93			
	H ₆	-177.02	-168.79	-160.30	5.10	5.29	6.78			
H ₇	-175.24	-166.79	-157.81	0.00	0.00	0.00				

Table 2

Response	Cov.	Orig.	2004- 2007 excl.	+ 2015- 2016 excl.	Orig.	2004- 2007 excl.	+ 2015- 2016 excl.	Orig.	2004- 2007 excl.	+ 2015- 2016 excl.	Orig.	2004- 2007 excl.	+ 2015- 2016 excl.
		Est.	Est.	Est.	Est. Error	Est. Error	Est. Error	LB	LB	LB	UB	UB	UB
95% Home range size	(Intercept)	0.68	0.70	0.72	0.08	0.09	0.09	0.53	0.54	0.55	0.85	0.89	0.90
	Kappa	7.83	7.84	7.82	0.10	0.10	0.11	7.64	7.64	7.60	8.02	8.05	8.04
	SOI _{1-yr} lag shape	0.03	0.03	0.03	0.02	0.02	0.02	0.00	0.00	0.00	0.06	0.06	0.07
Total distance travelled	(Intercept)	0.36	0.37	0.37	0.06	0.06	0.06	0.26	0.26	0.26	0.48	0.49	0.49
	Kappa	6.10	6.26	6.09	0.11	0.17	0.12	5.88	5.93	5.85	6.31	6.59	6.33
	SOI _{1-yr} lag	0.03	0.03	0.04	0.01	0.01	0.01	0.02	0.01	0.02	0.05	0.05	0.06
	IOD _{1-yr} lag	-0.97	-0.94	-1.10	0.36	0.36	0.40	-1.69	-1.71	-1.89	-0.27	-0.19	-0.31
	IOD _{2-yr} lag shape	2.68	-0.56	2.67	0.20	0.39	0.21	-1.33	-1.33	-1.33	0.19	0.19	0.19
Response	Cov.	Est.	Est.	Est.	SD	SD	SD	LB	LB	LB	UB	UB	UB
Mean bearing from colony	Intercept	-2.87	-2.92	-2.95	0.14	0.15	0.15	-3.16	-3.21	-3.24	-2.6	-2.63	-2.66
	Kappa	1.01	1.01	1.04	0.19	0.19	0.2	0.65	0.66	0.66	1.39	1.42	1.44
	IOD _{1-yr} lag	0.13	0.14	0.16	0.07	0.08	0.08	-0.01	-0.01	0.02	0.27	0.3	0.32
	SOI _{1-yr} lag	-0.24	-0.22	-0.23	0.1	0.1	0.1	-0.44	-0.42	-0.44	-0.05	-0.02	-0.05

Results

In general although your sample size extends over many years which is phenomenal in today's world, your sample size compared to colony size is small, how do you know these females are representative. This needs explaining including references to other studies.

We have included additional information in the 'Animal handling and instrumentation' section [L119-132] of the manuscript to clearly indicate the non-bias capture of females from the colony. While the sample size is small in some years, this due to the influence of weather on the timing and duration of fieldwork.

Previous work on the study population reported that females displayed considerable population-level repeatability/consistency in the total distance travelled during foraging trips (Speakman et al 2021).

While bearing and home range size were not analysed in the 2021 study, we feel that, given the random selection of females for capture, the results are representative of the population.

Discussion

I find the discussion is a bit repetitive of the introduction and of itself. I would like to see it broader given the topic of this manuscript. I would like to see a short discussion putting these results in context with both: a) other benthic Otariids (predominantly Galapagos / Australian / South American sea lions) particularly as they are all affected by SOI and; b) How difference and how big a shift is this relative to results already known for pelagic species? Given you say in the introduction and discussion that for pelagic species these effects have been documented. There is no discussion of either in here and they are needed.

As discussed in the paragraph of the discussion starting with 'While several studies have reported negative impacts of El Niño events on marine predators', comparisons between the impacts of the ENSO on AUFS and on other pinnipeds is quite difficult due to the influence of ENSO on SE Australian climate being vastly different from e.g. the Americas. Further, the differences in the levels of primary productivity in most regions and in Bass Strait (i.e., Bass Strait is a shallow region of low primary productivity while other studies have been conducted in regions of moderate-high primary productivity) could have substantial impacts on how ENSO influences the species.

With respect to benthic foraging otariids, no ENSO related effects have been reported for Australian sea lions, while Galapagos and South American sea lions experience different climate conditions under ENSO events. Additionally, many of the studies focus on pup production and body condition, not foraging behaviour or space use.

Due to the differences in ENSO influence on climate, the most suitable species to make comparisons with would be the Australian sea lion (see above), the New Zealand fur seal (pelagic forager) and New Zealand sea lion (no reported effects). Studies on NZ fur seals have focused on pup production and body condition (Bradshaw et al 2000, 2003; Boren et al 2006), and the interpretation of ENSO effects were inconsistent (e.g., Bradshaw et al 2003 referred to both La Nina and El Nino events as drivers of poor prey availability but also that La Nina was associated with good prey availability).

Nonetheless, where possible, we have attempted to improve the context in relation to other studies. [L328-338].

Figures and Tables

I think the current Tables 1 & 2 could go in supplementary data and that a biological data summary table needs to go in the paper i.e. number of females in each year and a mean of those years KHR, distance etc. And for Table S1 and its summary in the manuscript there the animals need to be listed in order by year – currently the first six or so years data are all jumbled by year.

Tables 1 and 2 have now been moved to the supplementary data and the summary table requested has been added (Table 1). Table S1 has been ordered by year.

Minor changes

Introduction **Ln 26** please define foraging effort in the context you are using here – you use it throughout to refer to home range used and time/distance from shore – however there are other description of foraging effort so you need to clearly define what you are calling foraging effort for this paper

In the discussion, we discuss the impacts of environmental change on other aspects of foraging effort (e.g., dive rate, proportion of time spent diving, etc). As such, I believe that defining foraging effort in the context of this paper alone (i.e., travel distance, space use) is inappropriate.

Pg 4 Ln 13 I think there should be an “as the” or “because the” in this sentence ie This is particularly concerning BECAUSE THE balance of phases of large-scale climate drives are predicted to change over the coming decades”

Amended. [L60]

Methods

How were the females chosen at random? 3400 pups and therefor females is a very large colony –

were the females always from one end of the colony? From the area of the colony most accessible? Evenly or randomly spaced throughout the colony?

Fieldwork was conducted for periods of 1-3 weeks, with the timing and duration of fieldwork (including the number of days of work while on the island) determined by weather and logistics.

In view of the stochasticity of female attendance, timing of fieldwork is random relative to when females are in attendance. In addition, while the colony may have an annual pup production of ~3400, outside of the mating season, females stagger their attendance patterns such that there is never that many females at the colony at the same time. Attendance on the island is driven by several factors (e.g., time ashore/at sea, pup/female condition, weather, prey availability). Furthermore, individual females shift their suckling locations on the island in relation to weather (e.g. amount of rainfall, wind direction etc.) which impacts their access for capture. Hence, given the numerous factors involved, we are confident that there was no bias in the animals selected for capture and that they represent a random sample of the pup-provisioning female population.

We have included additional information in the '*Animal handling and instrumentation*' section of the manuscript outlining the accessibility of breeding sites, movement of females relative to weather, and the random selection of females.

Pg 14 Ln 12 Please add a reference relating to the sentence that ends "...dependant pup."

Reference added. [L315]

Pg 18 sentence starting "Shifts in habitat use" please add a reference after lower body condition and at the end of the sentence

References added. [L430-431]

General for discussion

For Bass Strait, there is extensive commercial fishing allowed, particularly bottom trawling. Influences and changes of fishing intensity over this time just needs a mention as over the last 20 years I would be betting that fishing effort, methods., allowed catch etc may deplete prey at differing times but can also alter the benthos – and it may be worth mentioning as a variable that was not looked at but should be considered for a complete overall picture.

There are extensive south-east Australian trawl fisheries. However, as shown in Arnould & Kirkwood (2008) and Goldsworthy et al (2003) these occur over the shelf slope and beyond and, therefore, do not overlap with the foraging habitat of female Australian fur seals (see Arnould and Kirkwood 2008). While there are minor amounts of gillnet fishing and scallop fishing on the continental shelf, these are localised, minimal relative to the available habitat in Bass Strait.

Figure 1 Please make the colony blue circle – black it is very hard to pick out at the moment and impossible if the paper is printed in black and white (which is what I did)

Thank you for pointing this out. Colony now indicated by a black circle.